# Characterization of Age-Associated, Neutrophil-to-Lymphocyte Ratio (NLR) and Systemic Immune-Inflammatory Index (SII) as Biomarkers of Inflammation in Geriatric Patients with Cancer Treated with Immune Checkpoint Inhibitors: Impact on Efficacy and Survival

**DOI:** 10.3390/cancers15205052

**Published:** 2023-10-19

**Authors:** Khalil Choucair, Caroline Nebhan, Alessio Cortellini, Stijn Hentzen, Yinghong Wang, Cynthia Liu, Raffaele Giusti, Marco Filetti, Paolo Antonio Ascierto, Vito Vanella, Domenico Galetta, Annamaria Catino, Nour Al-Bzour, Azhar Saeed, Ludimila Cavalcante, Pamela Pizzutilo, Carlo Genova, Melissa Bersanelli, Sebastiano Buti, Douglas B. Johnson, Claudia Angela Maria Fulgenzi, David J. Pinato, Maluki Radford, Chul Kim, Abdul Rafeh Naqash, Anwaar Saeed

**Affiliations:** 1Barbara Ann Karmanos Cancer Institute, Department of Oncology, Wayne State University School of Medicine, Detroit, MI 48201, USA; choucairk@karmanos.org; 2Vanderbilt-Ingram Cancer Center, Nashville, TN 37232, USA; caroline.nebhan@gmail.com (C.N.); douglas.b.johnson@vumc.org (D.B.J.); 3Department of Surgery and Cancer, Hammersmith Hospital Campus, Imperial College London, London SW7 2BX, UK; alessiocortellini@gmail.com (A.C.); c.fulgenzi@unicampus.it (C.A.M.F.); david.pinato@imperial.ac.uk (D.J.P.); 4Operative Research Unit of Medical Oncology, Fondazione Policlinico Universitario Campus Bio-Medico, 00128 Rome, Italy; 5Department of Medicine, Kansas University Medical Center, Kansas City, KS 66211, USA; shentzen@kumc.edu (S.H.); gradford@uiowa.edu (M.R.); 6The University of Texas MD Anderson Cancer Center, Houston, TX 77030, USA; ywang59@mdanderson.org; 7Department of Internal Medicine, Baylor College of Medicine, Houston, TX 77030, USA; cynthia.x3.liu@kp.org; 8Medical Oncology Unit, Azienda Ospedaliero Universitaria Sant’Andrea, 00189 Rome, Italy; raffaelegiusti@yahoo.it; 9Phase 1 Unit, Fondazione Policlinico Universitario Agostino Gemelli, Istituto di Ricovero e Cura a Carattere Scientifico (IRCCS), 71013 Rome, Italy; marco.filetti@gmail.com; 10Istituto Nazionale Tumori IRCCS Fondazione Pascale, 80131 Napoli, Italy; paolo.ascierto@gmail.com (P.A.A.); vitovanella1@gmail.com (V.V.); 11Medical Thoracic Oncology Unit, IRCCS Istituto Tumori Giovanni Paolo II, 70124 Bari, Italy; galetta@oncologico.bari.it (D.G.); annamaria.catino@gmail.com (A.C.); pamela.pizzutilo@gmail.com (P.P.); 12UPMC Hillman Cancer Center, Department of Medicine, Division of Hematology and Oncology, University of Pittsburgh, Pittsburgh, PA 15232, USA; nnalbzour211@med.just.edu.jo; 13Department of Pathology and Laboratory Medicine, University of Vermont Medical Center, Burlington, VT 05401, USA; azhar.saeed@uvmhealth.org; 14Novant Health Cancer Institute, Charlotte, NC 28204, USA; 15UOC Clinica di Oncologia Medica, IRCCS Ospedale Policlinico San Martino, 16132 Genova, Italy; carlo.genova1985@gmail.com; 16Dipartimento di Medicina Interna e Specialità Mediche, Università degli Studi di Genova, 16126 Genova, Italy; 17Medical Oncology Unit, University Hospital of Parma, 43125 Parma, Italy; 18Medicine and Surgery Department, University of Parma, 43121 Parma, Italy; sebastiano.buti@unipr.it; 19Lombardi Comprehensive Cancer Center, Georgetown University, Washington, DC 20057, USA; chul.kim@gunet.georgetown.edu; 20Medical Oncology/Stephenson Cancer Center, University of Oklahoma, Oklahoma City, OK 73019, USA; abdulrafeh-naqash@ouhsc.edu

**Keywords:** immune checkpoint inhibitors, biomarkers, geriatric oncology, circulating inflammatory markers, systemic immune-inflammatory index

## Abstract

**Simple Summary:**

Older adults have a unique biology characterized by inflammation. Inflammation, however, is thought to affect the efficacy of immune therapies in patients with cancer, and there are currently no biomarkers to predict the response of older patients with cancer to immune checkpoint inhibitors (ICI). In this study, we explored the potential of two markers of inflammation (NLR: neutrophil-to-lymphocyte ratio, and SII: systemic immune-inflammatory index) in the blood to predict a response to ICI in patients ≥80 years old. We showed that lower levels of NLR and SII were significantly associated with a better response in patients ≥80 years, exclusively, and that low levels of SII were associated with prolonged survival independent of other clinicopathologic features. Pre-treatment levels of inflammatory markers are thus potential biomarkers that can be used to predict better response rates and survival in geriatric patients with cancer treated with ICIs.

**Abstract:**

Background: Geriatric patients (≥80 years) are underrepresented in immune checkpoint inhibitor (ICIs) clinical trials. However, their unique biology may affect their response to ICIs. There are currently no established biomarkers of the response to ICIs in adult patients with cancer that can help with patient selection. Methods: We built a multicenter, international retrospective study of 885 patients (<80 years: n = 417, 47.12%; ≥80 years: n = 468, 52.88%) with different tumor types treated with ICIs between 2011 and 2021 from 11 academic centers in the U.S. and Europe. The main outcome measures were objective response rates (ORR), progression-free survival (PFS) and overall survival (OS) stratified by age and circulating inflammatory levels (neutrophil-to-lymphocyte ratio (NLR) and systemic immune-inflammatory index (SII)). Results: Patients ≥80 years with low NLR (NLR-L) and SII (SII-L) had significantly higher ORR (vs. high NLR [NLR-H], *p* < 0.01 and SII-H, *p* < 0.05, respectively). At median follow-ups (13.03 months), and compared to SII-H, patients with SII-L had significantly longer median PFS and OS in patients <80 (*p* < 0.001), and ≥80 years (*p* < 0.001). SII-L was independently associated with longer PFS and OS (HR: 0.61 and 0.62, respectively, *p* < 0.01). Conclusion: Lower inflammation pre-ICI initiation may predict an improved response and survival in geriatric patients with cancer.

## 1. Introduction

Cancer is often a disease of aging, and immune checkpoint inhibitors (ICIs) have revolutionized cancer treatment [1]. The participation of older patients in ICI trials has been suboptimal, particularly at the extremes of age [2]. A recent evaluation of a large, multicenter cohort of patients over the age of 80 years with cancer suggests that ICIs have a favorable efficacy and toxicity profile in older adults [3]. However, up to two-thirds of patients treated with ICIs demonstrate resistance to these drugs, limiting their benefit while still exposing patients to risks of morbid or even life-threatening adverse events [4,5,6]. Tumor type is a major predictor of ICI efficacy, and some existing biomarkers, albeit imperfect, are currently used to select patients for ICI therapy (e.g., MSI-H, TMB or PD-L1 levels). The impact of tumor type and other markers, however, has been derived from major trials and studies where the participation of older patients has been minimal. None of the currently used markers have been exclusively studied in this population. This is particularly important in the setting of older and frailer patients with more comorbidity. There is thus an unmet need for biomarkers of response that can help with patient selection and monitoring responses to treatment in older patients with cancer.

Immunosenescence, the age-related remodeling process of the immune system, is thought to influence the efficacy of ICIs [7]. It results in immune dysregulation via both cellular and humoral immunity, with depletion of lymphocyte reserves, fewer CD4+ and CD8+ T cells, decreased variety of regulatory and memory T-cells and an overall increased pro-inflammatory state [8,9,10]. Chronic aging-related inflammation, “inflammaging”, is reflected in the higher serum levels of IL-6, CRP and TNF observed in adults ≥70 years old [11,12,13]. While common in older individuals, inflammaging is thought to be an evolutionary conserved mechanistic pillar of aging that is shared by age-related diseases, and is seen with some but not all individuals as we age [14]. Upon treatment with ICIs, a pro-inflammatory host state decreases the response to these therapies, as higher levels of pro-inflammatory markers in the tumor microenvironment (TME) [15,16] reduce tumor-infiltrating lymphocytes and confer worse survival [17,18,19]. We sought to explore the impact of pre-treatment levels of circulating inflammatory markers on response to ICIs in a cohort of patients 80 years (yr) of age or older (≥80 yr) in comparison to those younger than 80 (<80 yr). 

## 2. Material and Methods

### 2.1. Study Design and Data Sources

After institutional review board approval at each participating institution with a waiver of written informed consent, a multicenter international database of patients with different tumors treated with ICIs at 11 academic centers in the U.S. and Europe was built (Appendix A). Retrospective, de-identified data of 885 patients treated between 2011 and 2021 were collected consecutively, and patients were stratified by age at treatment initiation: <80 yr (n = 417) and ≥80 yr (n = 468). For the <80 yr cohort, specifically, and to avoid selection bias, the first consecutive 417 patients treated during this time period were selected and included. ICI agents included anti-programmed cell death protein-1/programmed death-ligand 1 (PD-1/PD-L1) or anti-cytotoxic T-lymphocyte-associated protein-4 (CTLA-4) therapies; chemotherapy/ICI combinations were excluded. Baseline characteristic data were collected, including age at treatment initiation, sex, performance status (Eastern Cooperative Oncology Group/ECOG), tumor histologies, stage at treatment initiation, prior lines of therapy, type of ICI therapy and time on therapy. 

Pre-treatment levels of circulating neutrophils, lymphocytes and platelets were collected to compute the neutrophil-to-lymphocyte ratio (NLR) and the systemic immune-inflammation index (SII) for each patient. NLR was defined as the absolute number of neutrophils divided by the absolute number of lymphocytes. SII is the product of platelet count and neutrophil count divided by lymphocyte count (SII = platelets × neutrophils/lymphocytes) (Appendix A). Optimal cut-off values for high (H) vs. low (L) levels were determined using receiver operating characteristic curves, as follows: NLR-L <3 (vs. ≥3 NLR-H) and SII-L (<600 vs. ≥600 SII-H). 

### 2.2. Outcomes and Statistical Analysis

Responses to treatment were reported in the database according to the evaluation of the locally treating oncologist based on the modified immune Response Evaluation Criteria in Solid Tumors (iRECIST, v1.1), and were categorized as progressive disease, stable disease, partial response (PR) and complete response (CR). Objective response rate (ORR) was defined as the proportion of patients who achieved PR and CR per the treating physician’s assessment. Chi-square (*χ*^2^) tests were used to compare categorical variables. Kaplan–Meier curves assessed progression-free survival (PFS) and overall survival (OS) for all patients and across age (<80 yr vs. ≥80 yr) and pre-treatment levels of inflammatory marker (H vs. L) groups. PFS was defined as the time from ICI initiation to progression or death or to the last follow-up for patients who were progression-free. OS was defined as the time from ICI initiation to death or to the last follow-up for patients who were still alive. The median follow up time was 13 months, and the cut-off date for the survival analysis was 22 June 2022; all patients who were progression-free or alive at the date of analysis were censored on that date. The prognostic value of each variable was evaluated with a multivariate Cox proportional hazard model. A *p*-value < 0.05 was considered statistically significant and, where applicable, a 95% confidence interval (CI) was provided. All statistical analyses were performed using Microsoft Excel (2007) and DATAtab Online statistics Calculator (2022; available at: https://datatab.net (Last accessed on 9 October 2023).

## 3. Results

### 3.1. Patient Characteristics

Table 1 summarizes patients’ characteristics. NSCLC was more common in patients <80 yr, while MEL was more common in patients ≥80 yr (*p* < 0.001). Median age at treatment initiation was 80, 65 and 82.8 years in the general, younger (<80 yr) and older (≥80 yr) cohorts, respectively (*p* < 0.001). Most patients had stage IV disease at ICI initiation (74.95%), and ICIs were either first-line (56.74%) or second-line (34.85%) therapy. Pembrolizumab was the most commonly used ICI (46.99%), in general and in patients ≥80 yr specifically compared to durvalumab in patients <80 yr. Patients <80 yr had better performance status (*p* < 0.001) and spent longer time on ICI therapy compared to older patients (*p* < 0.001). In patients <80 yr, 35.12% of patients had NLR-L and 64.88% had NLR-H, while 24.15% had SII-L and 75.85% had SII-H. In patients ≥80 yr, 38.62% of patients had NLR-L and 61.38% had NLR-H, while 33.72% had SII-L and 61.38% had SII-H. 

### 3.2. Clinical Outcomes

Patients <80 yr had higher ORR compared to those ≥80 yr, irrespective of the inflammatory markers levels (*p* < 0.001; Table 2). When stratified based on levels of circulating inflammatory markers, patients ≥80 yr with NLR-L had significantly higher ORR (vs. NLR-H; *p* < 0.01). There was no difference in ORR across NLR levels in patients <80 yr (Figure 1). Similarly, patients ≥80 yr with SII-L had significantly higher ORR (vs. SII-H; *p* < 0.05), with no difference in ORR across SII levels in patients <80 yr. 

Median follow-up time was 13.03 months (1.0–11.5 months). ***PFS:*** There was no significant difference in median PFS between both age groups. Median PFS was longer in patients <80 yr with NLR-L (19.63 months; 95% CI: 16.48–22.78; vs. NLR-H: 10.00 months; 95% CI: 7.90–12.10; *p* < 0.001), with no difference observed in patients ≥80 yr (*p* = 0.74) (Figure 2A). Compared to SII-H, patients with SII-L had a significantly longer median PFS in those <80 yr (20.40 vs. 11 months; *p* < 0.001) and ≥80 yr (8.77 vs. 5 months; *p* < 0.001) (Figure 2B). 

***OS:*** There was no significant difference in OS between both age groups. Compared to NLR-H, patients <80 yr and those ≥80 yr with NLR-L had longer median OS (24.07 and 11.57 vs. 14.00 and 11.57 months, respectively; *p* < 0.01) (Figure 3A). Similarly, SII-L was associated with longer median OS in patients <80 yr (27.33 months vs. 16.00 months; *p* < 0.001), and ≥80 yr (12.28 months vs. 7.00 months; *p* < 0.001) (Figure 3B). 

Results from the multivariate analysis for PFS and OS revealed that SII-L was independently associated with longer PFS and OS (HR: 0.61 and 0.62, respectively, *p* < 0.01; Table 3). Other significant factors include an age younger than 80 and good performance status (ECOG < 2). 

## 4. Discussion

Determinants of response to ICIs in older cancer patients remain poorly understood, and aging-associated biomarkers of ICI response in this population have not been explored before. Immunosenescence and inflammaging are inter-related biologic processes that underlie aging and carcinogenesis. They are hypothesized to share a common end-product, immune suppression [7,10,13], via the dysregulation of cellular and humoral immunity [9,10] and the modulation of the TME composition [16,17,18,19]. Here, low levels of NLR and SII were associated with significantly higher ORR in patients ≥80 yr but not in younger patients, suggesting that low levels of either of these markers can potentially serve as a biomarker of a response to ICI in patients ≥80 yr, exclusively. Our findings are supported by several studies across different tumor types treated with ICIs, whereby lower NLR and SII levels identified patients with a greater ORR to ICIs [20,21,22,23,24,25], and a decline in NLR levels post-ICI initiation, was associated with significantly improved ORR, PFS and OS [20,21]. However, none of these studies explored the potential use of these markers as discriminatory biomarkers for geriatric patients with cancer. To our best knowledge, this is the first study exploring inflammatory markers as potential age-associated predictors of response to ICIs. Of note, while lower NLR and SII were associated with better response rates in patients ≥80 yr, pre-treatment levels of these markers did not significantly differ by age. This can be explained by our selection of 80 yr as an age cut-off, whereby biological differences between younger patients at the upper range level (e.g., 79 years) and older ones at the lower range level (e.g., 80 years) may not be as pronounced, if existent. This could be addressed by comparing more distant age groups, such as ≤70 yr vs. ≥80 yr, for example. Nevertheless, the significant association between low NLR and SII levels and better ORR is likely a conservative estimation of this impact given the proximity of the two compared age groups as described earlier. 

The impact of low levels of circulating NLR and SII in patients ≥80 yr was also consistent for OS but did not consistently reflect on PFS with respect to age. This can be explained by a smaller sample size of patients ≥80 yr with NLR-L, for whom PFS data were available (n = 45). However, in a population of older patients with more comorbidity, achieving higher ORRs, irrespective of survival, carries clinical significance, as high response translates to better tumor control and, ultimately, an improved quality of life. The positive effect of SII on survival has been reported in NSCLC [22,26], RCC [27], CRC [28], breast [29] and gastric cancers [30], and across a variety of histologies [31]. In our study, low SII levels were an independent predictor of longer OS and PFS, irrespective of other variables, including tumor type. Younger age and better performance status were also independent predictors of longer OS and PFS, in keeping with the known impact of these two factors on survival in cancer patients. In a retrospective analysis of 444 patients with gastric cancer, Wang et al. reported that high SII levels were significantly associated with an unfavorable prognosis, independently predicted OS and were superior to NLR for predicting OS [30]. In a meta-analysis of nine studies involving 2441 patients with NSCLC, elevated SII indicated significantly poorer OS, PFS, time to recurrence (TTR) and cancer-specific survival (CSS), and SII showed a significantly higher prognostic value compared to NLR [22]. Similarly, in a retrospective analysis of 1383 patients with CRC, those with low SII levels following radical surgery had better OS and disease-free survival (DFS), and SII was an independent predictor of both OS and DFS [28]. This effect is consistent across several histologies, as a meta-analysis of 22 studies involving 7657 patients reported that high SII correlated with poor OS, TTR, DFS, PFS and CSS, irrespective of the cut-off value, sample size and patients’ ethnicity, thus highlighting high SII as a potential pan-cancer prognostic marker associated with poor outcomes [31]. 

A limitation of our study is the retrospective nature of the clinical database and potential site-related variations in data reporting, as well as selection bias. Similarly, the independent consecutive selection of patients in both subgroups (<80 vs. ≥80) did not allow for a strict control of potential age-related covariables. As such, a propensity-matching system could have been performed to reduce such bias and balance the impact of covariables across both subgroups. Also, the cut-off values for the inflammatory markers were determined internally according to the ROC analysis. These cut-offs were derived from a large sample size and are comparable to previously published studies, providing indirect validation of the ROC analysis [26,29,30]. Nevertheless, prospective studies are still needed to externally validate the selected values. Our clinical cohort is heterogeneous, reflecting different disease stages, histologies and therapies. For example, around 25% of patients had stages I-III of a disease compared to stage IV, and were included in the survival analysis. While this should be taken with caution while drawing conclusions, the fact that OS and PFS benefits based on SII status were observed in the multivariate analysis independent of the aforementioned co-variables suggests a solid pan-cancer prognostic value for SII. Last, and to account for biologic age differences across individual patients (rather than calendar age), it would be interesting in the future to directly explore the impact of NLR and SII (high vs. low), independent of age. 

## 5. Conclusions

In conclusion, the findings of this study support a potential role for low circulating NLR and SII levels in predicting a response to ICIs. While a large body of the literature supports our overall findings of a positive impact of lower levels of inflammatory markers on survival in patients treated with ICIs, our study is the first to explore this with respect to aging. Future studies are warranted to refine the predictive value of circulating inflammatory markers within specific histological cohorts.

## Figures and Tables

**Figure 1 cancers-15-05052-f001:**
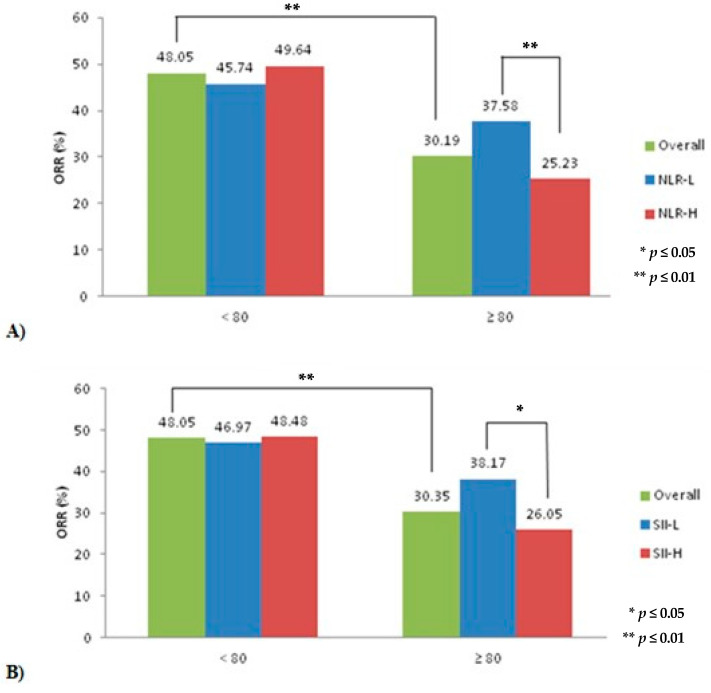
Treatment response to ICIs in patients <80 vs. ≥80 yr stratified by levels of pre-treatment circulating inflammatory markers. Response to ICI in patients <80 vs. ≥80 yr by (**A**) NLR level and (**B**) SII level. The overall column represents ORR by age group, irrespective of baseline inflammatory levels. ORR: Objective Response Rate (%). NLR: neutrophil-to-lymphocyte ratio. SII: systemic immune inflammatory index. H: high. L: low.

**Figure 2 cancers-15-05052-f002:**
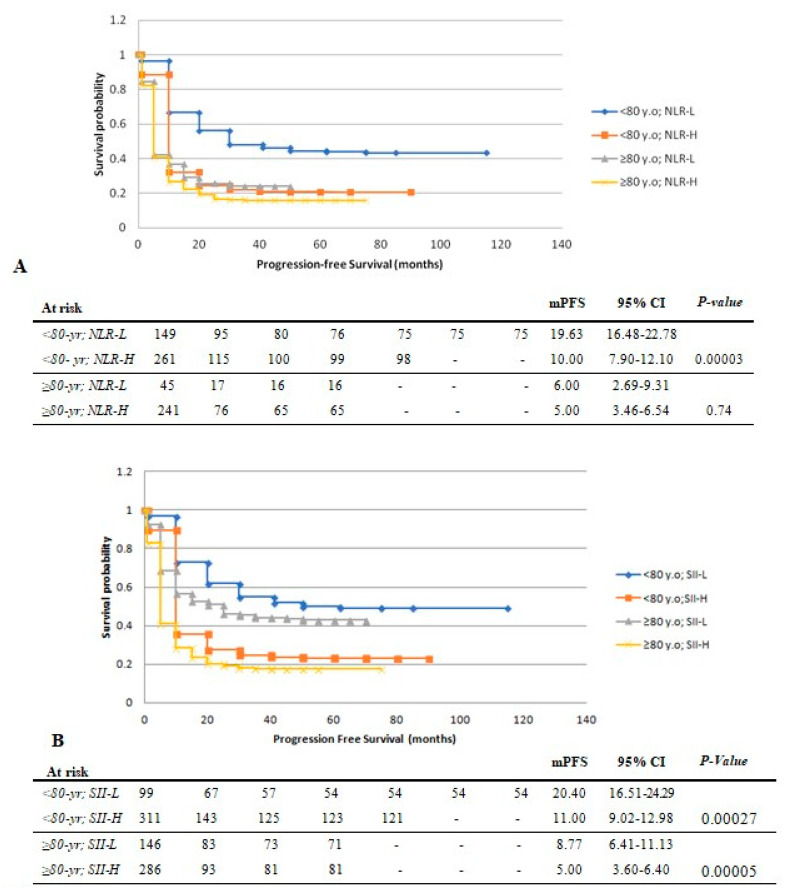
Kaplan–Meier plots of progression-free survival in patients <80 vs. ≥80 yr stratified by levels of pre-treatment circulating inflammatory markers. (**A**) NLR level. (**B**) SII levels. mPFS: median progression-free survival. NLR: neutrophil-to-lymphocyte ratio. SII: systemic immune inflammatory index. H: high. L: low. 95% CI: 95% confidence interval.

**Figure 3 cancers-15-05052-f003:**
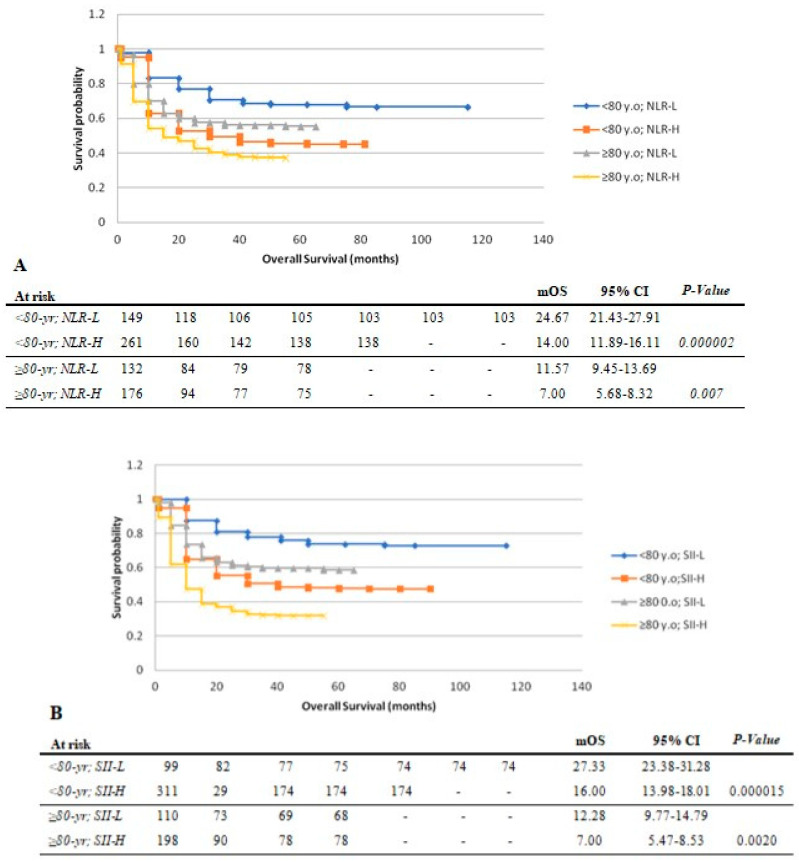
Kaplan–Meier plots of overall survival in patients <80 vs. ≥80 yr stratified by levels of pre-treatment circulating inflammatory markers. (**A**) NLR levels. (**B**) SII levels. mOS: median overall survival. NLR: neutrophil-to-lymphocyte ratio. SII: systemic immune inflammatory index. H: high. L: low. 95% CI: 95% confidence interval.

**Table 1 cancers-15-05052-t001:** Baseline characteristics of patients.

	<80 Years	≥80 Years	Overall	*p*-Value
Number of Patients (N)	417	468	885	-
Age (years) at ICI start, median (range)	65.0 (16.0–79.9)	82.8 (80.0–94.6)	80.0 (16.0–94.6)	<0.001
Sex (n = 697; %)				0.012
M	141/244 (57.8)	305/453 (67.3)	446/697 (64.0)	
F	103/244 (42.2)	148/453 (32.7)	251/697 (36.0)	
ECOG (n = 743; %)				<0.001
0–1	275/296 (92.9)	363/447 (81.2)	638/743 (85.87)	
2	20/296 (6.76)	70/447 (15.67)	90/743 (12.11)	
>2	1/296 (0.34)	14/447 (3.16)	15/743 (2.02)	
Tumor types, N (%)				<0.001
Melanoma	118 (28.3)	205 (43.80)	323 (36.5)	
NSCLC	282 (67.62)	220 (47)	502 (56.72)	
SCLC	15 (3.60)	2 (0.43)	17 (1.92)	
RCC	1 (0.24)	9 (1.92)	10 (1.13)	
Bladder/GU	1 (0.24)	21 (4.49)	22 (2.49)	
Tongue/Larynx/glottis	-	3 (0.64)	3 (0.34)	
Gastric/esophageal	-	3 (0.64)	3 (0.34)	
HCC	-	3 (0.64)	3 (0.34)	
Sarcoma	-	2 (0.44)	2 (0.22)	
Stage (n = 882; %)				0.123
I	7/414 (1.69)	10/468 (2.14)	17 (1.93)	
II	11/414 (2.90)	21/468 (4.49)	33 (3.74)	
IIIA	47/414 (11.35)	34/468 (7.26)	81 (9.18)	
IIIB/C	46/414 (11.11)	44/468 (9.4)	90 (10.2)	
IV	302/414 (72.95)	359/468 (76.71)	661 (74.95)	
Prior lines of therapy (n = 749; %)				0.028
0	176/296 (59.46)	249/453 (54.97)	425/749 (56.74)	
1	106/296 (35.81)	155/453 (34.22)	261/749 (34.85)	
2	9/296 (3.04)	30/453 (6.62)	39/749 (5.21)	
≥3	5/296 (1.69)	19/453 (4.19)	24/749 (3.20)	
ICI (n = 696; %)				<0.001
Pembrolizumab	100/244 (40.98)	227/452 (50.22)	327/696 (46.99)	
Ipilimumab	4/244 (1.64)	17/452 (3.76)	21/696 (3.02)	
Nivolumab	79/244 (32.38)	160/452 (35.40)	239/696 (34.34)	
Atezolizumab	15/244 (6.15)	19/452 (4.20)	34/696 (4.88)	
Avelumab	1/244 (0.41)	0	1/696 (0.14)	
Durvalumab	36/244 (14.75)	7/452 (1.55)	43/696 (6.18)	
Cemiplimab	0	3/452 (0.67)	3/696 (0.43)	
ICI combination	9/244 (3.69)	19/452 (4.20)	28/696 (4.02)	
Time (months) on ICI, median (range)				<0.001
(n = 697)	11.0 (1.0–82)	4.55 (1.0–61.4)	7.50 (1.0–82.0)	

ICI: Immune checkpoint inhibitor; M: male; F: female; ECOG: Eastern Cooperative Oncology Group; NSCLC: non-small cell lung carcinoma; SCLC: small-cell lung carcinoma; RCC: renal cell carcinoma; GU: genitourinary; HCC: hepatocellular carcinoma.

**Table 2 cancers-15-05052-t002:** Treatment response and survival.

	Overall (n = 885)	<80 Years (n = 417)	≥80 Years (n = 468)	
mOS (months; [95% CI])	12.88 [11.74–14.02]	14.30 [12.62–15.98]	12.80 [11.23–14.37]	
mPFS (months; [95% CI])	8.60 [7.52–9.68]	9.33 [7.73–10.93]	8.33 [6.84–9.82]	
				*p*-value
ORR, (n = 638), (%)	229/638 (35.89)	114/237 (48.1)	115/401 (28.68)	<0.001
CR	53 (8.31)	14 (5.91)	39 (9.73)	
PR	176 (27.59)	100 (42.19)	76 (18.95)	

mOS: median overall survival; mPFS: median progression-free survival; CR: complete response; PR: partial response; ORR: Objective response rate; 95% CI: 95% confidence interval.

**Table 3 cancers-15-05052-t003:** Multivariate analysis of Overall Survival (OS) and Progression-free Survival (PFS).

	Hazard Ratio (HR) for OS		Hazard Ratio (HR) for PFS	
Characteristic	HR (95% CI)	*p*-Value	HR (95% CI)	*p*-Value
Age at ICI initiation < 80 (vs. ≥80)	0.26 (0.19–0.33)	<0.001	0.4 (0.31–0.51)	<0.001
Sex: male (vs. female)	1.27 (1.01–1.61)	0.044	1.12 (0.9–1.4)	0.298
ECOG < 2 (vs. ≥2)	0.58 (0.44–0.77)	<0.001	0.65 (0.5–0.86)	0.002
Tumor type NSCLC (vs. MEL + others)	1.1 (0.2–8.0)	0.91	1.3 (0.3–5.0)	0.75
MEL (vs. NSCLC + others)	0.5 (0.1–3.8)	0.52	1.2 (0.30–4.9)	0.78
Others (vs. NSCLC + MEL)	1.8 (0.2–13.1)	0.56	1.9 (0.5–8.0)	0.37
Stage < IIIB (vs. IIIB-IV)	0.64 (0.46–0.89)	0.008	0.51 (0.37–0.71)	<0.001
ICI: anti-PD1/PDL1 (vs. non-PD1/L-1)	0.85 (0.57–1.26)	0.414	0.92 (0.62–1.38)	0.69
NLR ≤ 3 (vs. >3)	0.89 (0.63–1.24)	0.479	0.82 (0.6–1.14)	0.244
SII ≤ 600 (vs. >600)	0.62 (0.44–0.88)	0.008	0.61 (0.43–0.86)	0.004

The Cox proportional hazard model included age, sex, performance status (ECOG: Eastern Cooperative Oncology Group), tumor type, stage, choice of immune-checkpoint inhibitor (ICI) therapy, pre-treatment levels of circulating neutrophil-to-lymphocyte ratio (NLR) and systemic immune-inflammation index (SII). The hazard ratio (HR) of death (for OS) and death-or-progression (for PFS) are provided along with the 95% confidence interval (95% CI), and the corresponding *p*-value. NSCLC: non-small cell lung cancer; MEL: melanoma; Others: include all other tumor types described in Table 1.

## Data Availability

Data are available upon reasonable request.

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
