# Peer review of "Characterization of Age-Associated, Neutrophil-to-Lymphocyte Ratio (NLR) and Systemic Immune-Inflammatory Index (SII) as Biomarkers of Inflammation in Geriatric Patients with Cancer Treated with Immune Checkpoint Inhibitors: Impact on Efficacy and Survival"

_cancers, 2023, doi:10.3390/cancers15205052_

Round 1
Reviewer 1 Report
Many thanks for the opportunity to review this manuscript which examines the relationship between inflammatory biomarkers and survival and efficacy of immunotherapy in older patients with cancer.
I have a number of suggested edits which I hope improve the manuscript prior to publication.
Firstly, please rephrase the title to remove "inflammatory biomarkers" - I would expect to see IL-6, TNF, CRP levels etc if this was the case - rephrase to specifically state Neutrophil:Lymphocyte Ratio/SII as these are the markers that are evaluated.
Not all older adults have increased "inflammation" - inflammaging is ultimately a variable state where some older adults do have heightened low grade "sterile chronic inflammation" and others do not - some nuance is required here in the abstract/intro such as "Increased inflammation/inflammaging is seen with some, but not all, older individuals as we age".
The premise of the study is sound and the hypothesis is clearly stated. It should be commended that this is a multi-centre study.
Again, there should be a more nuanced discussion on inflammaging in the introduction - https://www.nature.com/articles/s41574-018-0059-4
Were all tumors considered ? How were individuals identified at each site ? Is it consecutive or convenience sampling ? Was it by hospital coding ? How does this influence the nature of the cohort recruited ?? Why consecutive patients for those <80 and not perform propensity matching ?
It is incorrect to say serum levels of neutrophils/lymphocytes as serum refers to what is left following centrifugation after the blood is centrifuged so either say circulating neutrophils/lymphocytes or blood levels of neutrophils/lymphocytes
Provide a reference for SII cut-offs.
Figures are poor quality and need a higher dpi/resolution
I have no criticisms of the main results as they are clearly presented
Author Response
We thank reviewer 1 for their time and thorough review of our work. A point-by-point rebuttal is attached herein, addressing the reviewer's comments.

Reviewer 2 Report
Comment:
This retrospective aimed to assess simple biological predictive factor for ICI efficacy and the relation of age. There are several concerns. Mainly it could not be state that NLR and SII are predictive factor for ICI efficacy as they could be only prognostic factor. A group of paired patients untreated with ICI may help to answer to this question.
Abstract: the number and percent of patient in the 2 groups may be indicate in the abstract
Introduction: tumor type is a major predictor for ICI efficacy and there are some biomarkers already use to select patient for ICI according to the tumor type (MSI, TMB, PD-L1 …).
Method: 1) the median follow-up is short
2) Pairing the patients from the 2 groups for the main characteristics could be considered as there is important discrepancy in the 2 populations
3) Of course, geriatric evaluation is not available in this retrospective study but this is lacking and a weakness of this study
Results:
1) What are the rates of NLR and SII low and high in the two groups?
2) PFS differences according the groups are not important and should not exclude patients for ICI treatment
3) Tumor type should be included in the multivariate analysis
4) In the text it seems that SII is the only factor resulting from the multivariate analysis, but other factors are also significant
Discussion:
1) several publications cited in the discussion are in favor of a prognostic role for NLR and SII even in patients that doesn’t receive ICI
2) The fact that a NLR and SII is not observed in younger patient in this study is conflicting with previous study that have showed a prognostic value younger patient
Conclusion: It could not state that NLR and SII are predictive factor for ICI efficacy
Author Response
We thank reviewer 2 for their time and thorough review of our work. A point-by-point rebuttal is attached herein, addressing the reviewer's comments.

Reviewer 3 Report
The paper is very well written. the research design, presentation are both excellent.
There are just one major issue (please include the ethnicity and comorbidities details in baseline table) and a few minor formatting issues throughou the paper such as the spacing e.g., line 146 after "Table 2)." Figure 1 is also not sharp enough, the * looks like dots. Please also ensure the same font/size used throughout the manuscript.
Author Response
We thank reviewer 3 for their time and thorough review of our work. A point-by-point rebuttal is attached herein, addressing the reviewer's comments.

Round 2
Reviewer 2 Report
The author have answer to the question